# Immediate Autogenous Bone Transplantation Using a Novel Kinetic Bioactive Screw 3D Design as a Dental Implant

**Carlos Aurelio Andreucci** [1], **Elza M. M. Fonseca** [2,*] and **Renato N. Jorge** [3]

[1] Ph.D. Engenharia Biomédica, Mechanical Engineering Department, Faculty of Engineering, University of Porto, 4200-465 Porto, Portugal

[2] LAETA, INEGI, ISEP, School of Engineering, Instituto Politécnico do Porto, R. Dr. António Bernardino de Almeida, 4249-015 Porto, Portugal

[3] LAETA, INEGI, Mechanical Engineering Department, Faculty of Engineering, University of Porto, Rua Dr. Roberto Frias, 712, 4200-465 Porto, Portugal

* Correspondence: elz@isep.ipp.pt

**Abstract:** The restoration of osseous defects is accomplished by bone grafts and bone substitutes, which are also called biomaterials. Autogenous grafts, which are derived from the same individual, can retain the viability of cells, mainly the osteoblasts and osteoprogenitor stem cells, and they do not lead to an immunologic response, which is known as the gold standard for bone grafts. There are both different techniques and devices that can be used to obtain bone grafts according to the needs of the patients, the location, and the size of the bone defect. Here, an innovative technique is presented in which the patient's own bone is removed from the trigone retromolar region of the mandible and is inserted into a dental alveolus after the extraction and immediate insertion of an innovative dental implant, the BKS. The first step of the technique creates the surgical alveolus; the second step perforates the BKS in the retromolar region, and shortly after, the BKS containing the bone to be grafted is removed; the third step screws the BKS bone that collects in the created surgical alveolus. Experimental studies have shown the feasibility and practicality of this new technique and the new dental implant model for autogenous transplants.

**Keywords:** bone grafting; bone transplantation; dental implants; biomedical technologies; immediate implant

## 1. Introduction

Bone loss can be mechanical, systemic, or a combination of the two. Osteopenia can occur as a process of aging, and its worsening can lead to osteoporosis, which is usually associated with systemic problems and localized pathologies such as periodontitis [1,2].

Bone grafts can be divided into four types: an autograft or autogenous transplant, which is derived from the same individual; an allograft or allogenous transplant, which is derived from the same species but different individuals; a xenograft, which is derived from different species, mainly of porcine or bovine origin; and alloplastic synthetic materials [1].

The mechanism of action of a bone graft can be either through osteogenesis, osteoinduction, osteoconduction, or a combination of these three processes. Osteogenesis refers to the development of new bone by cells within the graft. Osteoinduction is a chemical process in which molecules within a graft convert internal cells into osteoblastic cells, which in turn, form bone. Osteoconduction is a physical effect in which the graft matrix provides a scaffold that promotes cells from the outside of the graft to migrate and form new bone [3,4].

Autogenous grafts can retain the viability of the cells, mainly the osteoblasts and osteoprogenitor stem cells, and they do not lead to an immunologic response, which is known as the gold standard for bone grafts and meets the tissue engineering triad of signaling molecules, cells, and scaffold [5,6].

Autogenous bone grafts can be obtained either from intra-oral or extra-oral sources. Among the intra-oral sources are healing dental extraction wounds, bone from edentulous ridges, bone harvested from within the jaw using trephine burs, bone formed in the wounds, and bone obtained from the maxillary tuberosity, ramus, and mandibular symphysis [7]. The extra-oral sources are from the iliac crest, which provides cancellous bone marrow, the tibia, and the calvaria. Extra-oral sources can cause postoperative infection, increased patient expense, and difficulties in terms of finding donor material [8].

There are different techniques and devices to obtain bone grafts according to the patient's needs, the location, and the size of the bone defect. These are the bone scrapers, where the bone is removed by scraping action, rotary instruments such as burs or trephine, bone chisels, rongeur pliers, and piezoelectric devices [7,8]. There are advantages and disadvantages to every surgical procedure. Priority should be given to procedures that are simpler, less invasive, less likely to cause complications, and achieve goals in the shortest time.

In this paper, we present an experimental study in a synthetic human mandible, where the new biomechanical device, BKS [9–13], is used to remove bone, such as a rotatory instrument, from the mandible (retro-molar region), and immediately, without the manipulation of this bone graft, be introduced into a dental alveolus, created specifically for this study, filling the bone-deprived region with an autogenous bone while the BKS dental implant is screwed in place. With this technique, there is no need for the external manipulation of the bone graft, as the stabilization of the particulate bone is performed by the BKS dental implant itself.

## 2. Materials and Methods

The innovation of the new mechanical device, BKS, as seen in Figure 1, is in using the same properties of the screw, adding angles and cutting grooves at its apex, decreasing the area of contact with the perforated material, and thus, increasing the cutting pressure of a conventional screw. In addition to this benefit, the innovation introduced was the ability to store and compact the chips from drilling and inserting the BKS through its dual flutes by flowing the chips that meet at a common point in the BKS's inner volume and are pressed together. After the material where the new BKS has been applied fills the entire internal volume with its chips, compaction increases the density of the material transferring the torque exerted in the pressure area to the entire internal volume of the BKS.

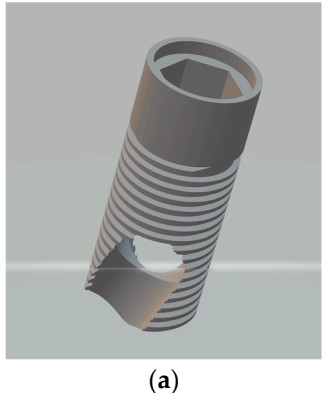
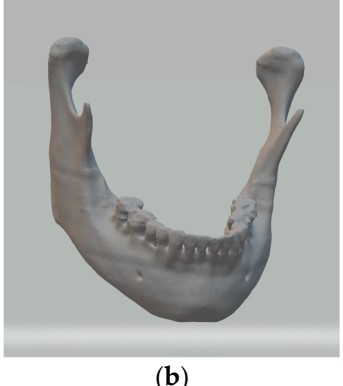

(**a**)           (**b**)

**Figure 1.** (**a**) BKS designed in SolidWorks®. (**b**) Mandible developed from DICOM original files converted into 3D CAD design.

For the execution of this work, the BKS dental implant models and the synthetic mandible were planned and developed in CAD program using SolidWorks® as seen in Figure 1, making it possible to obtain greater precision in the manufacture of the models and in the execution of the research proposed in this study.

For this experimental research, five mechanical devices of the new BKS model in the form of dental implants, made of commercially pure titanium, were machined (Usiform, Sao Carlos, Brazil) to be tested in human mandibles of synthetic bone (Synbones). A Techdrill Surgical Motor-1 with 150 Watts of power and a pre-set torque of 45 N/cm was used to make the drillings, simulating the tooth alveolus, removing the autogenous bone after insertion, and removing the new BKS device. The screw BKS implant into the tooth alveolus was newly created for this purpose, as seen in Figure 2.

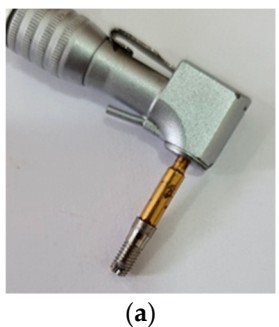
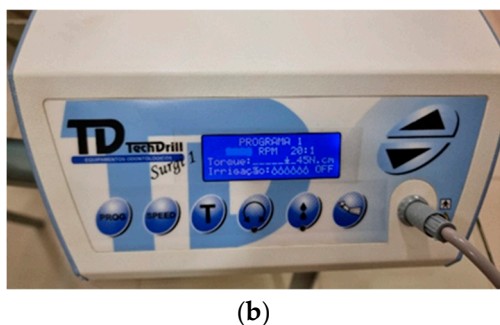

(**a**) (**b**)

**Figure 2.** (**a**) Contra-angle reducer 20:1 with adapter for implant insertion coupled into new BKS device in dental implant form. (**b**) A Techdrill Surgical Motor-1 with a pre-set torque of 45 N/cm.

As described in previous works [10], the BKS implant, when drilling into bone, can collect and compact the drilling chips, in this case, particulate bone, in its internal volume. In the technique proposed here, the BKS was removed after its insertion into the retromolar region of the mandible, to collect bone from this region (Figure 3) and graft it in (within BKS) in a surgical alveolus previously prepared for this purpose also in the same mandible (Figure 4). Low rotations were used to drill, insert, and remove the BKS (60 RPM) to keep the maximum amount of bone collected inside it. The technique described is characterized by the simplicity of the procedure, reduced contact of the collected material with the external environment, and faster surgical procedure, which requires the basic skills necessary for any dental surgeon.

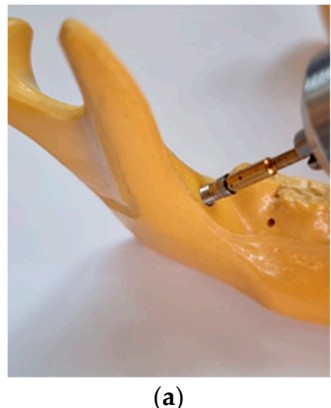
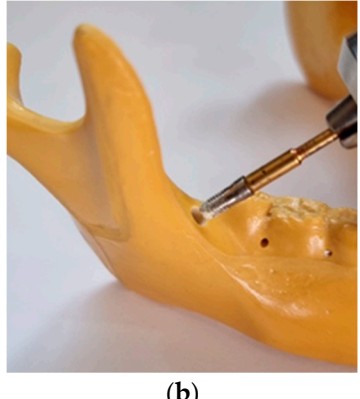

(**a**) (**b**)

**Figure 3.** (**a**) BKS insertion to collect the bone. (**b**) BKS removal with the bone chips inside its inner volume.

This simplified technique of autogenous bone transplantation for implants is only possible due to the inherent properties of the dental implant that are associated with the new biomechanical concept introduced by BKS, in which the bone that is normally discarded in the perforations is preserved and compacted inside, enabling its removal and reinsertion without direct contact with the transplanted material.

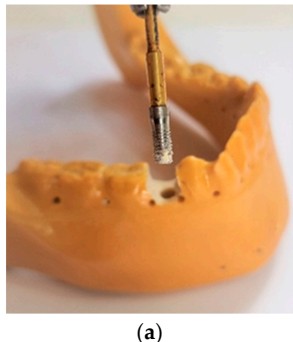 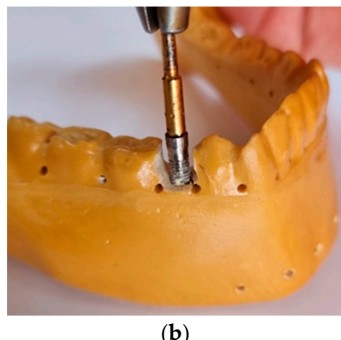

| (**a**) | (**b**) |

**Figure 4.** (**a**,**b**) BKS with the bone graft inside its inner volume implanted in the surgical alveolus created.

## 3. Results

Applying simple biomechanical concepts of bone drilling, screwing, biocompatibility, and bone implants, the engineering design of the BKS was created to stop the chip flow during bone drilling and screwing through a transfixing hole, creating a calculated material clog, maintaining, and compacting bone particles inside and through the new BKS device, as seen in Figure 5.

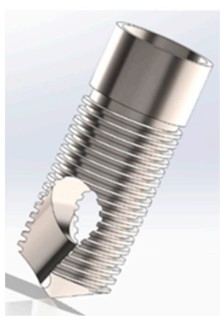

**Figure 5.** BKS biomechanism applied in this study.

The first step of the technique was to create the surgical alveolus using the measurements of the BKS (4 mm diameter by 10 mm length) in the region of tooth 45 (the right second pre-molar, as seen in Figure 6), using a standard drill in the same measurements as the BKS. The second step perforated the retromolar region with the new BKS device in its entirety and soon after, the interior volume of bone to be transplanted was removed, as seen in Figure 3. The third and last step was to screw the BKS with all the bone collected for grafting in its internal volume in the surgical alveolus created in the same mandible, as seen in Figure 4. There were always no intercurrences in the proposed procedures that were performed.

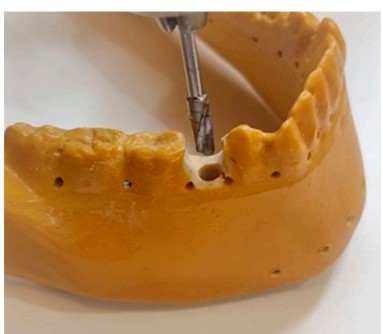

**Figure 6.** Surgical alveolus created in the region of the right second pre-molar (45).

## 4. Discussion

The types of autogenous bone grafts are cortical bone chips, osseous coagulum and bone blend, intraoral cancellous bone and marrow, extraoral cancellous bone and marrow, bone swagging, and autogenous bone blocks [1–5]. Osseous coagulum and bone blends are the types of autogenous grafts that are performed by BKS; in the experiment performed here, in synthetic bone, only the dried bone was grafted into the alveolus and stabilized by the BKS's screwing properties into the bone bed site. Especially in cases of recent dental extractions, with or without immediate rehabilitation with implants, the use of bone grafts is common and necessary to correct the bone defect or quality present, increasing the success rate of osseointegration, as well as solving the aesthetic and functional problems for future placement of a prosthesis [2,4,14].

The use of an appropriate method of bone repair is essential to achieve success in fractures or bone defects. Bone tissue engineering (BTE) has been shown to be another option for the restoration of bone defects. The main role of BTE is to promote bone regeneration by combining bone scaffolds with cells, growth factors, and bioactive agents. New techniques also include the production of living bone grafts in vitro by the long-term culture of cell-seeded biomaterials, regularly using bioreactors [2,15].

One of the goals of regenerative medicine is to biodegrade bone graft substitutes in the alveolus after extraction. In a study comparing the effects of a new bioresorbable nanostructured carbonated hydroxyapatite (CHA) with a bovine xenograft and a clot, CHA was shown to promote a higher rate of biodegradation and was a promising biomaterial for surgical alveolar preservation prior to implant treatment [16,17]. The accurate fitting and anchoring of bone grafts to the site are critical for the healing process. Standardized bone grafting matches the site to the graft and has been shown to be an effective treatment for small and medium-sized autogenous bone grafts with a low donor site morbidity and high graft success rates [1,18,19].

The simpler the method and the applicability of the bone graft, the greater the chances of success. The necessary amount of bone to be transplanted, the implant region, the function, and the aesthetics of the site will define the planning, the choice of techniques, and the type of graft to be used. Whatever the method applied, the association between the autogenous bone, BMP, and stem cells increases the chances of success in bone grafts [2,6]. The new BKS device can be used not only for bone transplantation in dental implants but also to obtain bone grafting to fill bone defects, where the amount of bone needed is close to the volume collected by this innovative device, or its content will be used to transport cells and molecules that are associated, which will make other types of biomaterials for transplants bioactive [2,6,14,20].

The idea with the tissue engineering scaffold is that you want to make a material that substitutes for the extracellular matrix on a temporary basis, allowing the cells to migrate, proliferate, differentiate, and attach to it, reestablishing their normal function, and secreting natural extracellular matrix [17]. It provides a synthetic. porous matrix that mimics the native extracellular matrix.

There are two types of scaffolds: natural, such as in skin, collagen and elastin (structural proteins), fibronectin and laminin (adhesive proteins), and proteoglycans mostly glycosaminoglycans (core proteins); in bone, collagen and hydroxyapatite (structural proteins and minerals); and synthetic scaffolds, ceramic and metal-based, responsible for vascular ingrowth and promoting bone repair (osteoconduction) [14,15,17].

Historical solutions have included filling in the resulting bone gaps with metal, animal bones, or pieces of bone from human donors, but none of these are optimal as they can cause infections or be rejected by the immune system, and they cannot conduct most of the functions of healthy bones [1,2]. Until now, the gold standard for bone tissue grafts has been homologous transplantation, in which a bone from the same individual is removed from one site to another [5].

Another ideal solution would be to grow a bone from the patient's own cells that are customized to the exact shape of the bone gap, and that is exactly what the authors

of [21] are currently testing. The method consists of the extraction of stem cells from a patient's adipose tissue and a CT scan to determine the exact dimensions of the bone defect to be treated. Then, a model with the precise shape of the bone is constructed, either with 3D printers or by carving decellularized bovine bones, leaving only the trabecular mineral lattice. The next step is to add the patient's stem cells to this lattice and place it in a bioreactor, a device that will simulate all the conditions that are found inside the body. Temperature, humidity, acidity, and nutrient composition all need to be precisely regulated for the stem cells to differentiate into osteoblasts and other cells, to colonize the mineral lattice, and remodel it with living tissue.

In the event of physical trauma, infection, or tooth loss, tissue regeneration and reprogramming are the only means of restoring homeostasis. As the body ages, the natural ability to regenerate tissues diminishes. Therefore, the need to understand this process has taken on considerable importance as the world's geriatric population continues to grow due to increased life expectancy. Tissue regeneration is a well-orchestrated molecular event in which each cell type plays its part. In multicellular organisms, the process has evolved redundancy, which is explained by reports that terminally differentiated cells can transdifferentiate into other cell types to meet the contingent needs of tissue regeneration. Metabolic conditions such as diabetes have a systemic effect on this process [22].

By using the technique proposed here, we decreased the chance of contamination, which has a high incidence in extraoral autogenous grafts [5,6,15], by decreasing the exposure time of the bone graft to the environment, and the manipulation of the grafted bone tissue is performed only indirectly by the BKS. The limitation of this technique occurs when it is necessary to fill more bone than can be collected by the new BKS device. By reducing the manual contact with the material to be transplanted, we automatically reduce the risk of contamination from the external environment, which in the case of the oral environment, can be saliva and especially bacteria, which can cause infection if they reach deep tissue levels. In addition to the handling time, an important factor in the risk of exposure to oral contaminants is the length of time that the bone graft is exposed to the environment for. With the new technique proposed here, the graft material remains in the BKS throughout the bone grafting procedure. It is immediately placed in the prepared surgical alveolus without direct manipulation.

The laboratory experiment proved the simplicity and efficiency of the proposed technique, and in vivo studies are being prepared to evaluate the advantages and disadvantages of the new BKS device in the form of a dental implant.

## 5. Conclusions

A new mechanical device, BKS, used as bone implants was used to create a new autogenous bone grafting technique. The control of mechanical factors during bone drilling, the safe management of the bone graft, and the maintenance of its stability, quality, and viability during drilling and implant screwing increase the chances of successful bone transplantation. This study elucidated the possibility of performing autogenous bone transplantation without the direct manipulation of the transplanted bone by using BKS and its drilling properties to collect the graft from the donor site and screwing at the surgical alveolus, simplifying the bone grafting technique for dental implants with the limitations described here.

Future studies will analyze the biological and clinical advantages of this simplified and innovative technique and its short-, medium-, and long-term benefits in dental implants. The addition of organic material to the in vivo study is expected to reduce the risk of contamination and loss of graft material during transplantation.

**Author Contributions:** Conceptualization, C.A.A.; methodology, C.A.A.; formal analysis, C.A.A.; investigation, C.A.A.; writing—original draft preparation, C.A.A.; writing—review and editing, E.M.M.F.; visualization, E.M.M.F.; supervision, R.N.J. All authors have read and agreed to the published version of the manuscript.

**Funding:** This research was funded by the Kuwait University, College of Engineering and Petroleum, Mechanical Engineering Department, grant number: EM04/22.

**Data Availability Statement:** Not applicable.

**Conflicts of Interest:** The authors declare no conflict of interest.

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
