# Peer review of "Immediate Autogenous Bone Transplantation Using a Novel Kinetic Bioactive Screw 3D Design as a Dental Implant"

_biomedinformatics, doi:10.3390/biomedinformatics3020020_

Round 1

Reviewer 1 Report

The paper deals with important topics in BioMedInformatics. The authors have presented an innovative technique is presented in which the patient’s own bone is removed from the trigone retromolar region of the mandible and inserted into a dental alveolus after extraction and immediate insertion of an innovative dental implant BKS.    

However, I have a number of suggestions:   

1. Abstract should be extended by the obtained results in step by step model.

2. I would suggest reinforcing the discussion section with the advantages of the proposed technique.

3.  In the Conclusions sections, added more data, about the accuracy rate you would like to achieve in future research.

Author Response

ANSWER – About the Reviewer’s comments, we are answering their concerns in the following lines.

The authors would like to acknowledge the Reviewer for their work on reading and suggesting improvements to the manuscript. We have addressed all the comments and the changes needed. Changes to the article were identified in green. We hope that the revisions in the manuscript and our accompanying answers will be sufficient to make our manuscript suitable for publication in Biomedinformatics.

Comments and Suggestions for Authors

The paper deals with important topics in BioMedInformatics. The authors have presented an innovative technique is presented in which the patient’s own bone is removed from the trigone retromolar region of the mandible and inserted into a dental alveolus after extraction and immediate insertion of an innovative dental implant BKS.

However, I have a number of suggestions:

1.Abstract should be extended by the obtained results in step by step model.

ANSWER – The abstract was extended by the obtained results in step-by-step model.

2.I would suggest reinforcing the discussion section with the advantages of the proposed technique.

ANSWER – We reinforced in the discussion the advantages of the proposed technique.

  1. In the Conclusions sections, added more data, about the accuracy rate you would like to achieve in future research.

ANSWER –More data about the accuracy was added to the conclusions.

Reviewer 2 Report

The manuscript is a brief report on the dental implant and BKS technique without elaboration on what BKS stands for in the entire text

I am unsure how an experiment using a synthetic mandible is employed as a way to demonstrate how bone tissue can be collected using the BKS method

The authors claim that “By using the technique proposed here we decreased the chance of infections that have a high incidence in extraoral autogenous grafts.”

How did they come to this conclusion by using the laboratory experiment?

I am afraid that the employed study design is questionable.

Author Response

ANSWER – About the Reviewer’s comments, we are answering their concerns in the following lines.

The authors would like to acknowledge the Reviewer for their work on reading and suggesting improvements to the manuscript. We have addressed all the comments and the changes needed. Changes to the article were identified in green. We hope that the revisions in the manuscript and our accompanying answers will be sufficient to make our manuscript suitable for publication in Biomedinformatics.

Comments and Suggestions for Authors

1.The manuscript is a brief report on the dental implant and BKS technique without elaboration on what BKS stands for in the entire text.

ANSWER – the authors thank the reviewer. We elaborated more about BKS in the text.

  1. I am unsure how an experiment using a synthetic mandible is employed as a way to demonstrate how bone tissue can be collected using the BKS method.

ANSWER –Bone grafting is performed using organic or inorganic material (hydroxyapatite) or a combination of both. Studies with synthetic bone particles have been routine in dentistry for decades, with robust literature on the subject. In this study, we are not analyzing the biological effects of BKS, but the ability of the proposed technique to transport synthetic bone-like material robustly proved and done in clinical practice, from one site to another with less manipulation. We are focusing on the technique of innovative BKS biomechanics.

  1. The authors claim that “By using the technique proposed here we decreased the chance of infections that have a high incidence in extraoral autogenous grafts.”

How did they come to this conclusion by using the laboratory experiment?

ANSWER –As no biological studies have been carried out at this time, the term "infection" has been corrected for “contamination” by materials in the environment. We also add more explanation about it in the discussion.

  1. I am afraid that the employed study design is questionable.

ANSWER –We clarified the questions in preview answers explaining that the study design is about the proposed BKS technique for the inorganic material, not biological responses. Also in the text: “The laboratory experiment proved the simplicity and efficiency of the proposed technique and in vivo studies are being prepared to evaluate the advantages and disadvantages of the new BKS device in the form of a dental implant”. We also added: “The addition of organic material to the in vivo study is expected to reduce the risk of contamination and loss of graft material during transplantation”.

Reviewer 3 Report

Being a brief report, I find the information to be adequate and of real interest to the dental surgeons involved in the field of implantology.

As a minor suggestion: please add manufacturer and location each time a material or device is being mentioned.

Please have the text checked for proper English and editing errors.

Author Response

About the Reviewer’s comments, we are answering their concerns in the following lines.

The authors would like again to acknowledge the Reviewer for their work on reading and suggesting improvements to the manuscript. We have addressed all the comments and the changes needed. Changes to the article were identified in blue.

Being a brief report, I find the information to be adequate and of real interest to the dental surgeons involved in the field of implantology.

  1. As a minor suggestion: please add manufacturer and location each time a material or device is being mentioned.

ANSWER – the authors thank the reviewer. We added the location and manufacturer, in a sentence after figure 1 (lines 87-88).

  1. Please have the text checked for proper English and editing errors.

ANSWER – the authors thank the reviewer. We made extension revisions to improve English and editing errors.

Reviewer 4 Report

Article not innovative but overall well structured with some critical issues especially related to the writing of the paragraphs to complete the work, a sign of the need for further study by the authors. In particular:

-check that all keywords are Pubmed MESH terms

- I find the introduction section too sparse and not well structured. Some general considerations on the possible reasons for bone loss, both linked to systemic and local pathologies linked to periodontitis, should be added in the initial part. In this regard, I ask you to insert in the reference section the following scientific work that could be of help to the reader:

Nesti M, Carli E, Giaquinto C, Rampon O, Nastasio S, Giuca MR. Correlation between viral load, plasma levels of CD4 - CD8 T lymphocytes and AIDS-related oral diseases: a multicenter study on 30 HIV+ children in the HAART era. J Biol Regul Homeost Agents. 2012 Jul-Sep;26(3):527-37. PMID: 23034272.

-line 52 lacks a description of the advantages and disadvantages of the various regenerative and bone augmentation techniques

-even the discussion section lacks scientificity. Some considerations, for example, should be added on the biological and cellular mechanisms underlying tissue regeneration and healing, with the description of the main pathways of biological signal transduction. In this regard, I suggest to insert in the reference section the following scientific work that could be of help to the reader:

Pagano S, Lombardo G, Caponi S, Costanzi E, Di Michele A, Bruscoli S, Xhimitiku I, Coniglio M, Valenti C, Mattarelli M, Rossi G, Cianetti S, Marinucci L. Bio-mechanical characterization of a CAD/CAM PMMA resin for digital removable prostheses. Dent Mater. 2021 Mar;37(3):e118-e130. doi: 10.1016/j.dental.2020.11.003. Epub 2020 Nov 27. PMID: 33257084.

-the presence of too many self-citations in references is detected. Remove repeated ones

Author Response

About the Reviewer’s comments, we are answering their concerns in the following lines.

The authors would like again to acknowledge the Reviewer for their work on reading and suggesting improvements to the manuscript. We have addressed all the comments and the changes needed. Changes to the article were identified in blue.

Comments and Suggestions for Authors

Article not innovative but overall well structured with some critical issues especially related to the writing of the paragraphs to complete the work, a sign of the need for further study by the authors. In particular:

  1. check that all keywords are Pubmed MESH terms

ANSWER – the authors thank the reviewer. We checked and changed the keywords.

  1. - I find the introduction section too sparse and not well structured. Some general considerations on the possible reasons for bone loss, both linked to systemic and local pathologies linked to periodontitis, should be added in the initial part. In this regard, I ask you to insert in the reference section the following scientific work that could be of help to the reader:

Nesti M, Carli E, Giaquinto C, Rampon O, Nastasio S, Giuca MR. Correlation between viral load, plasma levels of CD4 - CD8 T lymphocytes and AIDS-related oral diseases: a multicenter study on 30 HIV+ children in the HAART era. J Biol Regul Homeost Agents. 2012 Jul-Sep;26(3):527-37. PMID: 23034272.

ANSWER – the authors thank the reviewer. We made changes in the introduction section and bone loss was linked to systemic and local pathologies adding the proposed article.

  1. -line 52 lacks a description of the advantages and disadvantages of the various regenerative and bone augmentation techniques

-even the discussion section lacks scientificity. Some considerations, for example, should be added on the biological and cellular mechanisms underlying tissue regeneration and healing, with the description of the main pathways of biological signal transduction. In this regard, I suggest to insert in the reference section the following scientific work that could be of help to the reader:

Pagano S, Lombardo G, Caponi S, Costanzi E, Di Michele A, Bruscoli S, Xhimitiku I, Coniglio M, Valenti C, Mattarelli M, Rossi G, Cianetti S, Marinucci L. Bio-mechanical characterization of a CAD/CAM PMMA resin for digital removable prostheses. Dent Mater. 2021 Mar;37(3):e118-e130. doi: 10.1016/j.dental.2020.11.003. Epub 2020 Nov 27. PMID: 33257084.

ANSWER – the authors thank the reviewer. We added biological and cellular mechanisms underlying tissue regeneration and healing, with the description of the main pathways of biological signal transduction in the text. We also added the proposed article.

  1. -the presence of too many self-citations in references is detected. Remove repeated ones

ANSWER – the authors thank the reviewer. We removed the unnecessarily repeated self-citations.

Round 2

Reviewer 2 Report

The chosen study design does not support the hypothesis of the study.

Author Response

The authors would like again to acknowledge the Reviewer for their work on reading and suggesting improvements to the manuscript. We have addressed all the comments and the changes needed. Changes to the article were identified in blue. 

Reviewer 4 Report

All comments were added